# Recombination between the Fostera MLV-like Strain and the Strain Belonging to Lineage 1 of Porcine Reproductive and Respiratory Syndrome Virus in Korea

**DOI:** 10.3390/v14061153

**Published:** 2022-05-26

**Authors:** Go-Eun Shin, Ji-Young Park, Kyoung-Ki Lee, Bok-Kyung Ku, Choi-Kyu Park, Hye-Young Jeoung

**Affiliations:** 1Animal Disease Diagnostic Division, Animal and Plant Quarantine Agency, Gimcheon 39660, Korea; tlsrhdms924@korea.kr (G.-E.S.); jijipy@korea.kr (J.-Y.P.); naturelkk@korea.kr (K.-K.L.); kubk@korea.kr (B.-K.K.); 2College of Veterinary Medicine, Animal Disease Intervention Center, Kyungbuk National University, 80 Daehak-ro, Daegu 41566, Korea; parkck@knu.ac.kr

**Keywords:** porcine reproductive and respiratory syndrome virus (PRRSV), recombination, Fostera PRRS MLV, vaccine strain

## Abstract

Porcine reproductive and respiratory syndrome virus (PRRSV) is one of the most important pathogens in the swine industry worldwide. In Korea, Fostera PRRS commercial modified live virus (MLV) vaccines have been used since 2014 to control the PRRSV infection. In this study, two PRRSV-2 strains (20D160-1 and 21R2-63-1) were successfully isolated, and their complete genomic sequences were determined. Genetic analysis showed that the two isolates have recombination events between the P129-like strain derived from the Fostera PRRS MLV vaccine and the strain of lineage 1. The 20D160-1 indicated that partial ORF2 to partial ORF4 of the minor parental KNU-1902-like strain, which belongs to Korean lineage C (Kor C) of lineage 1, was inserted into the major parental P129-like strain. The 21R2-63-1 revealed that partial ORF1b of the P129-like strain was inserted into the backbone of the NADC30-like strain. This study is the first to report natural recombinant strains between Fostera PRRS MLV-like strain and the field strain in Korea. These results may have significant implications for MLV evolution and the understanding of PRRSV genetic diversity, while highlighting the need for continuous surveillance of PRRSV.

## 1. Introduction

Porcine reproductive and respiratory syndrome virus (PRRSV) belongs to the family *Arteriviridae* in the order *Nidovirales*, and is an enveloped, single-stranded, positive-sense RNA virus [1]. PRRS was first recognized in the United States in 1987 and isolated in Europe in 1990 [2,3]. Following the initial reports, several outbreaks have been widely documented, and the virus is now endemic to most swine-producing regions of the world [4]. The viral genome is approximately 15 kb in length and encodes 10 open reading frames (ORF): ORF1a, ORF1b, ORF2a, ORF2b, and ORFs 3–7, including ORF5 and ORF5a. [5]. ORF1a and ORF1b encode two large polyproteins that generate 14 non-structural proteins (nsp) [6]. Eight structural genes (ORF2a, ORF2b, ORF3–7, and ORF5a) encode structural proteins, including glycoprotein (GP) 2, small envelope protein (E), GP3, GP4, GP5, membrane protein (M), nucleocapsid (N), and ORF5a proteins, respectively [7,8]. PRRSV is divided into two types, PRRSV-1 (European) and PRRSV-2 (North American), which vary in approximately 40% in their nucleotide sequences [4]. The ORF5 sequence has been widely used to analyze the genetic variation and the molecular epidemiology of PRRSV [9]. Global phylogeny using the ORF5 sequence is characterized by many groups (subtypes 1–4 for PRRSV-1 and lineages 1–9 for PRRSV-2) within each genotype, which are composed of genetically and geographically distinct PRRSV [10,11].

Since the first identification of PRRSV-1 and PRRSV-2, in 2005 and 1994, respectively, both viruses have co-circulated in Korea [12,13]. Recent reports show that Korean PRRSV-2 belongs to lineages 1, 5, and 8, and Korean lineages (Kor A, B, and C), and the majority of Korean PRRSV-2 belong to lineage 5 [14]. However, a recent study showed that the PRRSV-2 lineage 1 population increased from 2014 (1.8%) to 2019 (29.6%) in Korea owing to the spread of sublineage 1.8 (NADC30-like viruses), comprising the second-largest population after lineage 5 (31.1%) in 2019 [14]. Since 1996, four commercial PRRS modified live virus (MLV) vaccines have been available in Korea for controlling PRRSV infections. The commercial PRRS MLV vaccines include the PRRSV-1 based Unistrain PRRS^®^ (Hipra, Amer, Girona, Spain), Porcilis PRRS^®^ (MSD Animal Health, Summit, NJ, USA), the PRRSV-2 based Ingelvac^®^ PRRS MLV (Boehringer Ingelheim Vetmedica Inc., St. Joseph, MI, USA), and Fostera^®^ PRRS MLV (Zoetis Animal Health US, Florham Park, NJ, USA) [15,16].

The recombination between wild-type and vaccine strains has increased along with the widespread use of PRRSV vaccines [17,18,19,20,21,22], and a possible recombination event between two PRRS MLV strains has been identified [23]. In particular, a recombination event between the Fostera PRRS MLV strain belonging to lineage 8 and the NADC30-like strain belonging to lineage 1 has been reported in the United States [24]. Recombination plays an important role in viral epidemiology and results in the genetic diversification of PRRSV [25]. The importance of recombination in PRRSV has been highlighted by the emergence of recombinant highly-pathogenic PRRSV (HP-PRRSV) in China [26]. After the first emergence of HP-PRRSV in 2006, the second major HP-PRRSV outbreak reported in 2009 and 2010 was associated with recombination events among different HP-PRRSV strains [27]. Many recombinant viruses of lineages 8 and 3 have been documented since 2010 in China [28,29].

In the present study, we isolated two PRRSV strains from pig farms, determined their complete genomic sequences, and analyzed novel recombinants to investigate the viral evolution.

## 2. Materials and Methods

### 2.1. Samples Collection and PRRSV Detection

From 2020 to 2021, lung samples of pigs with respiratory distress and atrophy accompanied by increased mortality were submitted to the Diagnostic Division of the Animal and Plant Quarantine Agency (APQA). In 2020, the 20D160-1 sample was collected from a pig farm infected with PRRSV in Chungcheong Province, which showed a growth rate of 95%, due to unexplained atrophy. In this farm, the use of the Ingelvac PRRS MLV vaccine (Boehringer Ingelheim Vetmedica) but non-use of Fostera PRRS MLV vaccine (Zoetis) was recorded. The 21R2-63-1 sample was obtained from a farm in Gyeongsang Province in 2021, which used the Fostera PRRS MLV vaccine (Zoetis) from 2020. Approximately 15% of weaned pigs in the herd experienced severe atrophy, accompanied by abdominal breathing and coughing. Total RNA was extracted from the samples using an RNeasy Mini Kit (Qiagen, Hilden, Germany), according to the manufacturer’s instructions. The 20D160-1 and 21R2-63-1 samples were identified as positive for PRRSV-2 using commercial VDX^®^ PRRSV HP MP RT-PCR and NA/EU Typing Nested PCR (Median Diagnostics, Gangwon, Korea).

### 2.2. Virus Isolation

MARC-145 cells were grown in Dulbecco’s Modified Eagle Medium (Gibco, Grand Island, NY, USA) supplemented with 10% fetal bovine serum (Gibco, Grand Island, NY, USA) and antibiotic-antimycotic solutions (Gibco, Grand Island, NY, USA) at 37 °C under a humid 5% CO_2_ atmosphere. The PRRSV isolation was conducted independently in separate rooms to exclude any contamination during virus adaptation. After incubation for 3 days, samples with visible cytopathic effects (CPE) in more than 80% of cells were passaged for further analysis. Total RNA was extracted from PRRSV-infected cellular suspensions (passage 3; P3) using an RNeasy Mini Kit (Qiagen, Hilden, Germany).

### 2.3. Sequencing Analysis

The cDNA was synthesized with gene-specific primers using a PrimeScript 1st Strand cDNA synthesis kit (Takara, Dalian, China), according to the manufacturer’s instructions. Polymerase chain reaction (PCR) was conducted to amplify eight overlapping fragments in the PRRSV full genome. The primers used in this study were based on a previous study, with some modifications [30,31,32]. The PCR products were sequenced in both directions using a commercial sequencing service company (Macrogen, Daejeon, Korea). The complete genome sequences of 20D160-1 and 21R2-63-1 were deposited in GenBank under the accession numbers OM681585 and OM681586, respectively. To identify the evolutionary relationship of the isolates, 7 global representative PRRSV strains (VR2332, P129, TJ, CG, SDSU73, NADC30, and NADC31) and 4 historical PRRSV strains from Korea (LMY, JB15-N-P31-GB, CA-2, and KNU-1902) in GenBank were utilized for sequence alignments, using the BioEdit software (Ibis Biosciences, Carlsbad, CA, USA).

### 2.4. Recombination Analysis

Recombination was detected using three methods [26]. First, genome sequences were scanned for possible recombination events and break points, by conducting a similarity plot analysis, using the Simplot software with a window size of 200 bp and a step size of 20 bp [33]. Second, the recombination detection program 4 (RDP4) was used to search for statistically significant recombination breakpoints, by setting seven recombination detection methods in the default settings [34]. Recombination events were only considered significant (*p* value ≤ 1 × 10^−6^) when supported by at least five of the seven detection methods. Finally, putative recombination events were supported by phylogenetic analysis of parental regions, using the maximum likelihood method, with 1000 bootstrapping values in MEGA-X (The Biodesign Institute) [35].

## 3. Results

### 3.1. Full Sequence Analysis

Two PRRSV strains (20D160-1 and 21R2-63-1) were successfully isolated from MARC-145 cells within three passages, and the full genome of each isolate was sequenced. The nucleotide sequences of ORF2–7 in the original sample (P0) and P3 virus of the two isolates shared 99.9% identity with a nucleotide difference (data not shown). The 20D160-1 and 21R2-63-1 isolates appeared clustered into lineages 8 and 1, respectively, based on their ORF5 gene homology. The complete genome alignments of 20D160-1 revealed that ORF1a, ORF1b, ORF5, ORF6, and ORF7 shared 97.97–99.62% nucleotide homology with the P129 strain (Fostera MLV strain), which was higher than the homology shared with the other reference strains (Appendix A). Specifically, ORF2–4 of 20D160-1 shared 95.15–96.75% nucleotide homology with the KNU-1902 strain belonging to the Kor C of lineage 1, but 82.72–88.44% nucleotide homology with the P129 strain. The ORF1a, ORF1b, and ORF2–7 of the 21R2-63-1 isolate shared 90.00–95.65% nucleotide homology with the NADC30 strain, which was higher than the nucleotide homology with other representative strains. The relationship of these strains with Korean historical strains showed a similar homology to other global reference strains (Appendix A).

### 3.2. Amino Acid Analysis of nsp2

The nsp2 gene of the 21R2-63-1 isolate had a discontinuous 131 amino acid (aa) deletion (111 + 1 + 19-aa), when compared with that of the VR2332 strain, and the deletion pattern was consistent with the NADC30 strain (Figure 1). In addition, there was a 6-aa deletion in the nsp2 gene of 20D160-1, which is identical to that of the P129 strain (Figure 1).

### 3.3. Recombination Analysis

Recombination events within the complete genomes of 20D160-1 and 21R2-63-1 isolates were detected using three different analytical methods. First, the Simplot analysis revealed that 20D160-1 is the recombination between the P129-like strain and the KNU-1902-like strain of Kor C in lineage 1 (Figure 2A). In addition, 21R2-63-1 was the result of the recombination between the P129-like strain and the NADC30-like strain (Figure 3A). Second, all seven methods in the RDP4 program were utilized to detect recombination events and breakpoints, and the results are summarized in Appendix A. The 20D160-1 and 21R2-63-1 had a remarkably high degree of certainty, with a *p* value < 1 × 10^−6^ in at least seven algorithms. Two recombination breakpoints within 20D160-1 were identified, which were located at nt 11,990 and nt 13,563, corresponding to nt 103 of ORF2 and nt 508 of ORF4. The results indicated that partial ORF2 to partial ORF4 of the minor parental KNU-1902-like strain was inserted into the major parental P129-like strain. Subsequently, two recombination breakpoints were identified at positions 7654 and 8599 in the 21R2-63-1, corresponding to nt 144 and nt 1089 in the nsp9 within ORF1b. Finally, 20D160-1 and 21R2-63-1 were differentially located on the phylogenetic tree of their parental regions (Figure 2B,C and Figure 3B,C). The minor parental region of 20D160-1 was clustered close to the KNU-1902 strain, but away from the lineages formed by the P129 strain (Figure 2B). In contrast, the major parental region of the 20D160-1 was clustered with the P129 strain in lineage 8 (Figure 2C). Although the minor parental region of 21R2-63-1 was closely related with the P129 strain, the isolate clustered with the NADC30 strain based on the major parental region (Figure 3B,C).

## 4. Discussion

Among the measures used to control PRRSV infection, vaccination with MLV vaccines has been the primary choice for the majority of pig farms. In 1996, the Ingelvac PRRS MLV vaccine (Boehringer Ingelheim Vetmedica) was introduced to control PRRS in Korea [15]. The Ingelvac PRRS MLV vaccine virus and its parental strain VR-2332 belong to lineage 5. The PRRSV-2 vaccine-like strains have become highly prevalent since the introduction of Ingelvac PRRS MLV vaccine (Boehringer Ingelheim Vetmedica), and the majority of Korean PRRSV-2 field isolates belong to lineage 5 [11,32,36]. In addition, a recombinant virus between the Ingelvac PRRS MLV strain and the Korean field strain was identified under the circumstances of Ingelvac PRRS MLV vaccine (Boehringer Ingelheim Vetmedica) being widely used in Korean pig herds [26]. Fostera PRRS MLV vaccine (Zoetis), which is based on the virulent US PRRSV isolate P129 of lineage 8, was registered in Korea in 2014 [37]. Previous studies demonstrated that lineage 8, derived from the Fostera PRRS MLV vaccine strain, was first detected in 2015, and was consistently detected at a low rate in Korea until recently [14,38]. In this study, the 20D160-1 belonging to lineage 8 shared 97.97% sequence homology with the P129 strain based on ORF5. The farm that collected 20D160-1 did not use the Fostera PRRS MLV vaccine (Zoetis), but from the detection of the P129-like strain it can be inferred that the Fostera PRRS MLV-like strain circulates in Korean farms.

In general, the nsp2 coding region is genetically the most variable region in the PRRSV genome, with notable substitutions, deletions, and insertions [39]. In this study, the 21R2-63-1 presented a deletion pattern (111-1-19-aa) consistent with that of the NADC30 strain when compared to the reference strain VR-2332, which is identical to the discontinuous deletion in nsp2 with several isolates of the virulent lineage 1 [40,41]. However, several studies have demonstrated that individual nsp2 deletions are not essential for virulence, suggesting that multiple factors can induce viral pathogenicity [42]. Previous studies demonstrated that nsp2 of PRRSV was divided into five groups based on insertion and deletion patterns, the “111-1-19-aa”, “100-aa”, “1-29-aa” deletion groups, “36-aa” insertion group and no insertion or deletion group represented in the VR-2332 strain. Interestingly, while the “111-1-19-aa” deletion has been previously reported to follow a lineage 1 deletion pattern, it also existed in some lineage 3 or lineage 8 strains. The inconsistency between the nsp2 patterns and ORF5 lineages might have resulted from the recombination of PRRSV [43]. Further functional studies on nsp2 deletion patterns might contribute to a deeper understanding of PRRSV pathogenicity, virus evolution, and genetic classification.

All PRRSV within the order *Nidovirales* use a mechanism of discontinuous transcription to generate an extensive set of subgenomic (sg) mRNAs during infection [44]. The transcription-regulating sequences (TRS) are located at or near the 5′ end of each structural protein-coding region (ORF2–7). TRS can form a stem-loop structure to generate sgRNA interactions with a conserved TRS sequence located at the 3′ terminus of the 5′ untranslated region (UTR) [45,46]. This discontinuous sgRNA transcription strategy is similar to the copy choice mechanism that is a widely accepted mechanism for PRRSV recombination. Therefore, a discontinuous sgRNA transcription strategy is a possible explanation for the high recombination frequency of PRRSV [26]. However, the recombination breakpoints were found to be random throughout the PRRSV genome, with an unclear role of TRS in recombination [47]. In this study, the start of the breakpoint of 20D160-1 (nt 11,990) was situated independently of the TRS region (UGAACC at 11,860–11,865 for sgmRNA2), while the end of the 20D160-1 breakpoint (nt 13,563) was located near the TRS region (UUAGCC at 13,555–13,560 for sgmRNA5). Further studies are needed to investigate the functional relationship between the TRS region and the recombination breakpoint.

Since the NADC30-like strain has recently become prevalent in China, the main Chinese outbreak patterns were recombinations of lineage 1 and MLV strains [19,20,48]. Interestingly, the NADC30-like virus showed characteristics of easy recombination with other strains [49]. After the NADC30 strain was found in China, it spread to most areas and continuously recombined with Chinese strains, resulting in at least five different patterns of recombination [26,48,50]. Recently, the lineage 1 population increased with the continual spread of the NADC30-like strain in Korea [14]. Therefore, Korean PRRSV strains belonging to the NADC30-like group, such as 21R2-63-1, have the potential for high-frequency recombination with the MLV strain. Interestingly, previous studies have shown that most recombination breakpoints were located in nsp2 or nsp9 of a recombinant between NADC30-like strains and other strains in China [45,51]. In the present study, the breakpoint of 21R2-63-1 was located in nsp9, indicating that nsp9 is also a hotspot for recombination events in Korea.

The minor parental strain of 20D160-1 was the KNU-1902 strain. KNU-1902 results from the recombination between the CA-2-like strain belonging to Kor C of lineage 1 and the KU-N1606-like strain belonging to Kor A. The region from ORF2 to ORF7 in the CA-2-like strain was introduced into the backbone of the KU-N1606-like strain [42]. In this study, the RDP4 analysis results of 20D160-1 indicated that ORF2 to ORF4 of the minor parental KNU-1902-like strain was inserted into the major parental P129-like strain. Therefore, the minor parental region of 20D160-1 is related to the CA-2-like strain. Interestingly, 20D160-1 was obtained from a farm in Chungcheong Province, which is the same region that KNU-1902 was collected from. Although the recombination pathway of the CA-2-like strain could not be identified, it can be inferred that the locally circulating CA-2-like strain caused recombination of 20D160-1.

The presence of a vaccine strain leads to variant PRRSV strains, due to recombination between the wild-type and vaccine strains [17,18,19,20,21,22]. In this study, we successfully isolated two natural recombinant viruses between the Fostera PRRS MLV strain and the Korean field strain, which were circulating in a commercial pig herd. Several studies have shown that new strains of PRRSV, resulting from MLV evolution or the emergence of recombination events, exhibit higher pathogenicity or virulence in pigs [52]. Therefore, it is necessary to further analyze the pathogenicity of the isolates identified in this study. In this study, it can be inferred that the continuous use of Fostera PRRS MLV vaccine (Zoetis) in the farm that collected 21R2-63-1 increased the possibility of recombination with the field PRRSV strain. Although the pig farm where the 20D160-1 was obtained does not use a Fostera PRRS MLV vaccination to control PRRS, our analyses predicted that the vaccine virus existed in the pig herds and could evolve through genomic recombination with the field virus. This suggestion can explain the continuous emergence of novel PRRSV strains and the expansion of PRRSV’s genetic diversity. Thus, the commercial MLV vaccination should be carefully considered in stable PRRSV-infected pig farms, and long-term use of PRRSV MLV should not be practiced in unstable PRRSV-infected pig farms. Additionally, continuous monitoring of the emergence of new PRRSV recombinants is necessary for the genetic characterization and evolutionary analysis of PRRSV in the future.

## Figures and Tables

**Figure 1 viruses-14-01153-f001:**
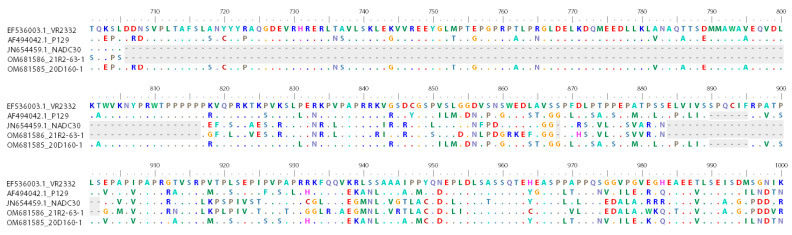
Analysis and comparison of amino acid in nsp2 compared to reference strains using BioEdit software. In comparison to VR2332, 131-aa deletions (111 + 1 + 19-aa) are highlighted in dark gray.

**Figure 2 viruses-14-01153-f002:**
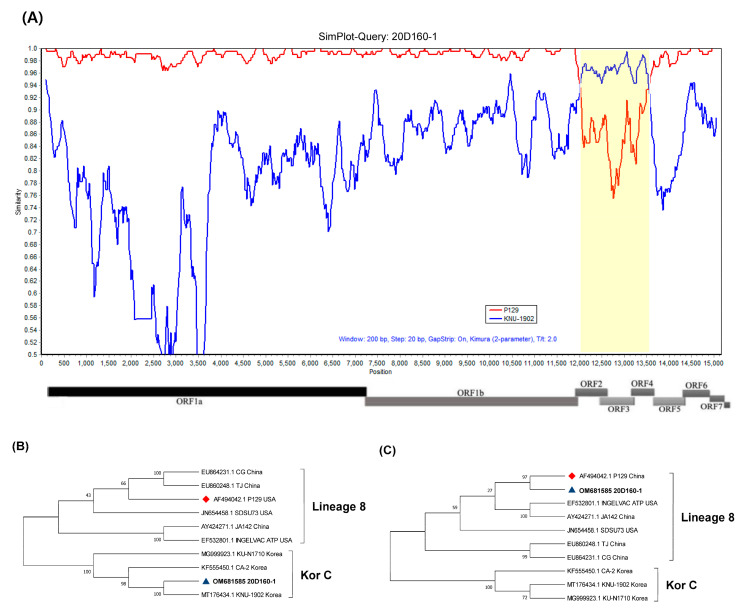
Recombination analyses of 20D160-1 and representative PRRSV lineages. (**A**) Similarity plot and bootscan analyses of 20D160-1 by Simplot. Yellow indicates the recombination region between 11,990 and 13,563, which covers ORF2, ORF3, and ORF4. The *y*-axis indicates the percentage similarity between the parental sequences and the query sequences. Phylogenetic trees of minor (**B**) and major (**C**) regions. The minor parental region of 20D160-1 was found to be related to KNU-1902-like strains, but the major parental region of 20D160-1 was closely related to the P129 strain. 20D160-1 is indicated by a blue triangle (▲), and vaccine strains are indicated by a red diamond (◆).

**Figure 3 viruses-14-01153-f003:**
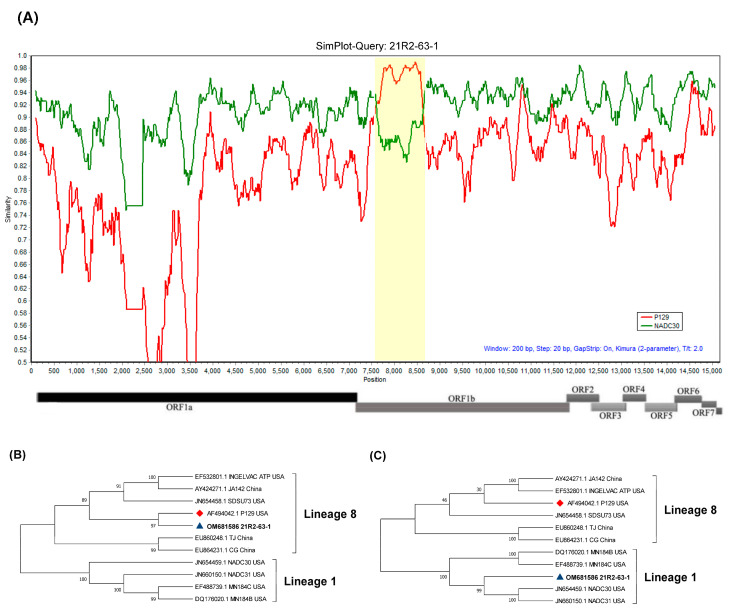
Recombination analyses of 21R2-63-1 and representative PRRSV lineages. (**A**) Similarity plot and bootscan analyses of 21R2-63-1 using Simplot. Yellow indicates the recombination region between 7654 and 8599 within ORF1b. The *y*-axis indicates the percentage similarity between the parental sequences and the query sequences. Phylogenetic trees of minor (**B**) and major (**C**) regions. The minor parental region of 21R2-63-1 was found to be related to the P129 strain, but the major parental region of 21R2-63-1 was closely related to NADC30 strain. 21R2-63-1 is indicated by a blue triangle (▲), and vaccine strains are indicated by a red diamond (◆).

## Data Availability

The sequences reported in the present manuscript have been deposited in the GenBank database under accession number OM681585 and OM681586.

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
