# Peer review of "Recombination between the Fostera MLV-like Strain and the Strain Belonging to Lineage 1 of Porcine Reproductive and Respiratory Syndrome Virus in Korea"

_viruses, 2022, doi:10.3390/v14061153_

Round 1

Reviewer 1 Report

The manuscript describes the isolation and characterization of two PRRSV strains obtained from Korean pig farms, determining their complete genomic sequences, and analyzing novel recombinants to study the evolution of PRRSV. The results showed that the strains (20D160-1 and 21R2-63-1) clustered into two different PRRSV strains (8 and 1, respectively), based on the homology of the ORF5 gene. The study is providing further evidence for the occurrence of events of recombination between one specific MLV vaccine strain and PRRSV field isolates. Although this finding is not original and those recombination events have already been demonstrated previously, the manuscript brings new evidence that improves the understanding of PRRSV evolution, so I think it should be published. .

Despite making an important contribution, it must be improved. The text of the abstract is confusing, preventing a better understanding of the whole study. I was only able to understand the main message after reading the complete manuscript. I also suggest that the authors change the title, as the probable recombination of two strains was observed and not between a strain and a lineage. There are also lots of acronyms and not necessary detailed strain identifications. A better text would be needed to highlight the main findings of the study.

All the remaining manuscript text looks ok, but an English grammar review would be most welcome. In addition, I suggest removing less informative data (such as those reported in Tables 1 and 2; transferring them to supplementary material) and keeping only the essential data, as the article is classified as a Communication.  

Author Response

Response to Reviewer 1 Comments

Point 1: The text of the abstract is confusing, preventing a better understanding of the whole study.

Response 1: Thank you for your comment. We have altered the sentence on lines 12, 13, 14, 16, 17, and 19 to be clearer.

Point 2: I also suggest that the authors change the title, as the probable recombination of two strains was observed and not between a strain and a lineage.

Response 2: We agree with your opinion. As suggested, we changed “Recombination between the Fostera MLV-like strain and the lineage 1 of porcine reproductive and respiratory syndrome virus in Korea” to “Recombination between the Fostera MLV-like strain and the strain belonging to Lineage 1 of Porcine Reproductive and Respiratory Syndrome Virus in Korea”.

Additionally, we changed “the lineage 1” to “the strain of lineage 1” on line 19.

Point 3: There are also lots of acronyms and not necessary detailed strain identifications. A better text would be needed to highlight the main findings of the study.

Response 3: Thank you for noting this oversight. The sentence now reads, “the 5’ untranslated region (UTR)” on lines 235 and 236. In addition, we have deleted “(Go-Eun Shin)” on line 289. If you have any further suggestion, please let me know.

Point 4: In addition, I suggest removing less informative data (such as those reported in Tables 1 and 2; transferring them to supplementary material) and keeping only the essential data, as the article is classified as a Communication.

Response 4: As suggested, we changed the table, accordingly. “Table 1 -> Supplementary Table 1” and “Table 2 -> Supplementary Table 2”

  • We added the word “(Supplementary Table 1)” on lines 133, 139, and 140.
  • On lines 162 and 177: “(Table 2)” -> “(Supplementary Table 2)”

Reviewer 2 Report

I reviewed the manuscript entitled “Recombination between the Fostera MLV-like Strain and the Lineage 1 of Porcine Reproductive and Respiratory Syndrome Virus in Korea”. In this study authors characterize an isolate of PRRSV recovered in Korea and demonstrate the existence of recombination events between this isolate and the vaccine strain.

Overall, I consider that it is an important study based on the relevance of PRRSV for the porcine industry. Although, the existence of recombination events between vaccine strains and field strains has been previously documented, I think is important to have scientific records to evidence the magnitude of this problem around the world.

I think the results presented in this study are well supported by different bioinformatic methods. I like the fact that is study is presented in a format of communication, that I think is perfect for the nature of this study.

At this point, I don’t have concerns to recommend the publication of this study. Some suggestion to improve the quality of this study, would be the inclusion of additional information regarding the epidemiology of this disease in Korea, highlighting information regarding the recombinant lineage presented herein. the addition of some details about clinical information would be important to point out the relevance of this recombinant virus.  Also, as a reader I would appreciate the inclusion of a phylogenetic tree to emphasize the results of section 3.1 and see the relationship of this recombinant virus with historical sequences from Korea.  I am not sure if this format of publication may allow the inclusion of an additional figure, otherwise it may be included as a supplementary figure.

Author Response

Response to Reviewer 2 Comments

Point 1: Some suggestions to improve the quality of this study, would be the inclusion of additional information regarding the epidemiology of this disease in Korea, highlighting information regarding the recombinant lineage presented herein.

Response 1: Thank you for your kind comment. We have added information, as follows: “Recent reports show that Korean PRRSV-2 belongs to lineage 1, 5, and 8, and Korean lineages (Kor A, B, and C), and the majority of Korean PRRSV-2 belong to lineage 5 [32, 40]. However, a recent study showed that the PRRSV-2 lineage 1 population increased from 2014 (1.8%) to 2019 (29.6%) in Korea owing to the spread of sublineage 1.8 (NADC30-like viruses), comprising the second-largest population after lineage 5 (31.1%) in 2019 [40]” on lines 49–54.

In addition, we changed “Korean lineage (Kor) C” to “Kor C” on lines 134 and 135.

Point 2: The addition of some details about clinical information would be important to point out the relevance of this recombinant virus.

Response 2: We agree with your opinion. To inform readers of an important point, we added clinical information on lines 80, 81, 84, and 85.

Point 3: Also, as a reader, I would appreciate the inclusion of a phylogenetic tree to emphasize the results of section 3.1 and see the relationship of this recombinant virus with historical sequences from Korea. I am not sure if this format of publication may allow the inclusion of an additional figure, otherwise, it may be included as a supplementary figure.

Response 3: Reviewer 1 suggested that removing less informative data such as table1 and table 2 transfer to supplementary material. Therefore, instead of the inclusion of a phylogenetic tree to see the relationship of this recombinant virus with historical sequences from Korea, we added homology with the Korean historical strains to supplementary Table 1. In addition, we added information related to the relationship between Korean historical strains and recombinant strains on lines 108–110 and 138–139. If you have any further suggestions, please let me know.
